# What Not to Overlook in the Management of Patients with Type 2 Diabetes Mellitus: The Nephrological and Hepatological Perspectives

**DOI:** 10.3390/ijms25147728

**Published:** 2024-07-15

**Authors:** Carlo Maria Alfieri, Paolo Molinari, Felice Cinque, Simone Vettoretti, Annalisa Cespiati, Daniela Bignamini, Luca Nardelli, Anna Ludovica Fracanzani, Giuseppe Castellano, Rosa Lombardi

**Affiliations:** 1Department of Nephrology, Dialysis and Renal Transplantation, Fondazione IRCCS Ca’ Granda Ospedale Policlinico, 20122 Milan, Italyluca.nardelli@unimi.it (L.N.); giuseppe.castellano@unimi.it (G.C.); 2Department of Clinical Sciences and Community Health, University of Milan, 20122 Milan, Italy; 3Post-Graduate School of Specialization in Nephrology, University of Milan, 20122 Milan, Italy; 4SC Medicina Indirizzo Metabolico, Fondazione IRCCS Ca’ Granda Ospedale Maggiore Policlinico di Milano, 20122 Milan, Italy; annalisa.cespiati@unimi.it (A.C.); daniela.bignamini@policlinico.mi.it (D.B.); anna.fracanzani@unimi.it (A.L.F.); rosa.lombardi@unimi.it (R.L.); 5Department of Pathophysiology and Transplantation, University of Milan, 20122 Milan, Italy

**Keywords:** chronic kidney disease, diabetic kidney disease, metabolic dysfunction-associated steatotic liver disease, liver fibrosis, hyperfiltration

## Abstract

Diabetes mellitus (DM) significantly impacts renal and hepatic function, necessitating comprehensive understanding and management strategies. Renal involvement, namely diabetic kidney disease (DKD), presents a global challenge, with increasing prevalence paralleling DM rates. Lifestyle modifications and pharmacotherapy targeting hypertension and glycemic control have pivotal roles in DKD management. Concurrently, hepatic involvement in DM, characterized by metabolic dysfunction-associated steatotic liver disease (MASLD), presents a bidirectional relationship. DM exacerbates MASLD progression, while MASLD predisposes to DM development and worsens glycemic control. Screening for MASLD in DM patients is of high importance, utilizing non-invasive methods like ultrasound and fibrosis scores. Lifestyle modifications, such as weight loss and a Mediterranean diet, mitigate MASLD progression. Promising pharmacotherapies, like SGLT2 inhibitors and GLP-1 agonists, demonstrate efficacy in both DM and MASLD management. Special populations, such as diabetic individuals undergoing hemodialysis or kidney transplant recipients, demand special care due to unique clinical features. Similarly, DM exacerbates complications in MASLD patients, elevating the risks of hepatic decompensation and hepatocellular carcinoma. Recognizing the interconnectedness of DM, renal, and hepatic diseases underscores the need for multidisciplinary approaches for optimal patient outcomes. The present review aims to present the main characteristics and crucial points not to be overlooked regarding the renal and hepatic involvement in DM patients focusing on the inter-relationships between the renal and the hepatic involvements.

## 1. Introduction

Type 2 diabetes mellitus (T2DM) is a chronic, multifactorial, and multisystemic disease that has reached epidemic proportions [1]. In fact, in 2019, approximately 463 million people were affected by diabetes, 90% of whom had type 2 diabetes mellitus, and this number is expected to increase in the coming years, especially due to the widespread adoption of a Western lifestyle [2].

Hyperglycemia is the basis of the micro- and macrovascular complications typical of diabetic disease, which significantly impact the quality of life of people with diabetes mellitus with increasing morbidity and mortality [3].

Diabetic pathology can involve and dysfunction numerous organs and systems, significantly impacting the survival and organ complications of affected patients [4]. Indeed, along with vascular involvement, whose main clinical manifestations are cardiovascular and kidney disease, type 2 diabetes can also affect the liver, exposing patients to an increased risk of hepatic steatosis and fibrosis, possibly evolving into cirrhosis and hepatocellular carcinoma [5]. Therefore, early and multidisciplinary assessment is of fundamental importance for patients with this condition in order to control possible risk factors and prevent disease evolution [6].

The objective of this review is to present the main characteristics and crucial points not to be overlooked regarding the renal and hepatic involvement of diabetes mellitus. In particular, this work will be divided into two parts, with the first part focusing on nephrological issues and the second part on hepatological issues. Common conclusions will be presented in the final part of this review.

## 2. Diabetes, Overview of Pathophysiology and Treatment

Diabetes mellitus is defined by the occurrence of fasting plasmatic glucose levels of more than 126 mg/dL in more than two determinations and/or glycated hemoglobin > 6.5% and/or plasmatic glucose levels of more than 200 mg/dL in a random determination or after an oral glucose load test (Figure 1) [7]. The main forms of diabetes are type 1 and 2, characterized by specific pathogenetic and clinical features, even though other forms have been described.

By this definition, from 2001 to 2009, youth with type 1 diabetes (T1DM) in the U.S. increased by 21% [8], whereas 95% of the 30 million people with diabetes have type 2 diabetes (T2DM). Risk factors include older age, race/ethnicity, male sex, and socio-economic status. Type 2 diabetes is rising in youth, especially among racial minorities. Global prevalence varies, with high rates in East Asia, South Asia, Australia, North America, and the Caribbean [9].

T1DM is characterized by the autoimmune destruction of pancreatic beta cells, resulting in absolute insulin deficiency. Its clinical presentation often includes polyuria, polydipsia, weight loss, and fatigue, commonly appearing in children and young adults. Rapid onset of symptoms necessitates prompt diagnosis and insulin therapy initiation to prevent ketoacidosis [7].

Differently from T1DM, which is characterized by absolute insulin deficiency, defective insulin secretion is central to T2DM, with insulin secretion failing to adequately compensate for insulin resistance, resulting in a low disposition index [7]. First-phase insulin secretion and responses to glucose are markedly impaired, leading to worsening hyperglycemia due to β-cell deterioration [10]. Clinically, it manifests with a spectrum of symptoms, including increased thirst, frequent urination, fatigue, and blurred vision. However, it often develops insidiously, with many individuals remaining asymptomatic for years. The progression of T2DM is characterized by worsening insulin resistance and declining pancreatic beta-cell function, leading to persistent hyperglycemia. Over time, untreated or poorly managed T2DM can result in complications such as cardiovascular disease, neuropathy, nephropathy, and retinopathy [11].

As for T1DM, enteroviral infections, altered gut microbiome, food timing (cereal, gluten), low vitamin D, perinatal factors, and nitrosamine exposure are risk factors [11,12,13]. Conversely, metabolic alterations are known risk factors for insulin resistance and the onset of T2DM. Among them, obesity is a major risk, as ectopic fat in the liver, muscle, and pancreas worsens insulin resistance and β-cell function [14]. Interestingly, different BMI thresholds apply to various populations, with Asians affected at lower BMIs [15]. Accordingly, weight loss improves overall insulin sensitivity also by reducing pancreatic fat. Also, sleep duration impacts obesity and diabetes risk, with obstructive sleep apnea contributing to both conditions [16].

Indeed, both T1DM and T2DM have largely distinct genetic bases. Higher T1DM prevalence in relatives suggests genetic risk, with human leukocyte antigen (HLA) variants contributing 50–60% of this risk (HLA-I, B*5701, and B*3906) [17]. Approximately 50 other genes also play smaller roles, affecting immune regulation, viral response, and endocrine function. Together, these genes account for about 80% of T1DM heritability [18]. Moreover, genome-wide studies have identified over 130 variants for T2DM, though these explain less than 15% of heritability [19]. Some variants, such as those in KCNQ1, show strong parent-of-origin effects [20] and factors like gene–gene interactions and epigenetics may account for the rest.

In addition, the onset of T2DM may be a consequence of some medications, namely drug-induced diabetes, such as glucocorticoids, antipsychotics (olanzapine), and immunosuppressants (calcineurin inhibitors), which can impair insulin secretion or increase insulin resistance [21,22]. Moreover, immune checkpoint inhibitors, such as PD-1/PD-L1 inhibitors, can induce autoimmune diabetes, manifesting as rapid-onset insulin-dependent diabetes mellitus. This condition is characterized by the destruction of pancreatic beta cells, leading to severe insulin deficiency [23].

Finally, a subtype of T2DM is gestational diabetes mellitus (GDM), which is defined as any degree of glucose intolerance with onset or first recognition during pregnancy. Although most cases resolve with delivery, the definition is applied whether or not the condition persisted after pregnancy and does not exclude the possibility that unrecognized glucose intolerance may have antedated or begun concomitantly with the pregnancy [7].

Independently of the cause of diabetes, management of the disease relies on the modification of lifestyle with weight loss and therapeutical strategies that have accumulated over time, with particular interests in new molecules, namely glucagon-like peptide receptor agonist (GLP1A) and sodium–glucose cotransporter 2 inhibitors (SGLT2i), as described in the next paragraphs [24,25].

An updated overview of patient management is shown in Figure 1.

**Figure 1 ijms-25-07728-f001:**
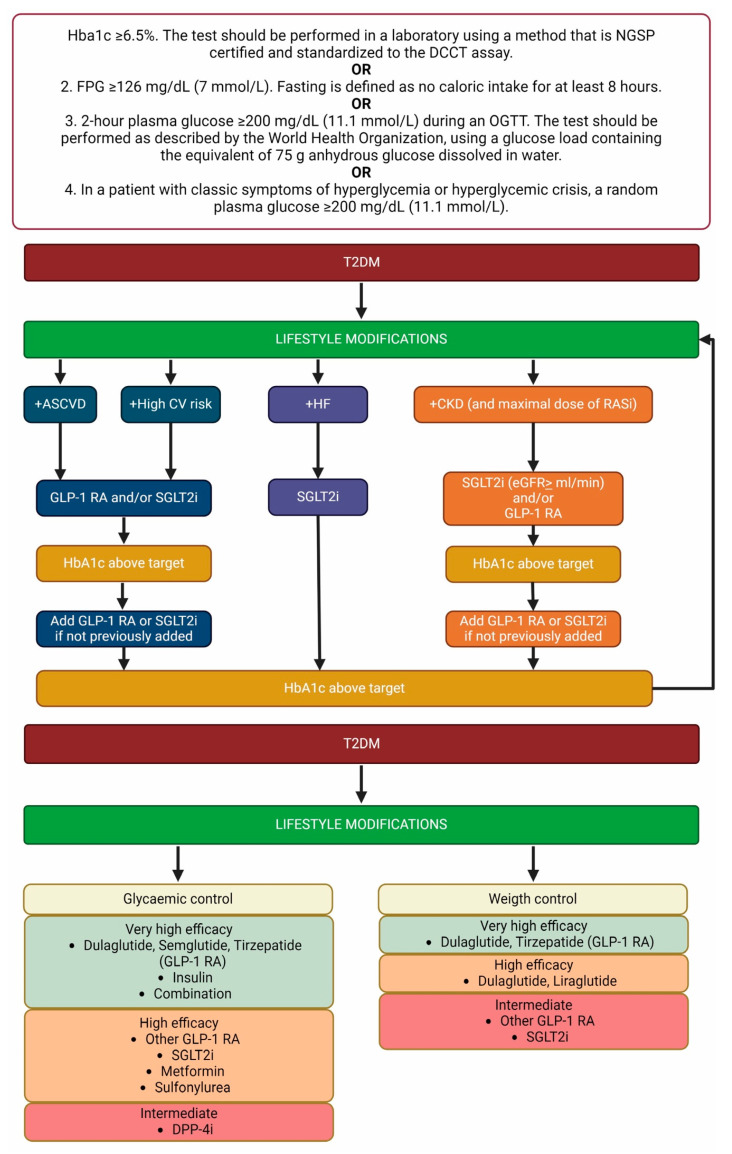
Overview of diabetes diagnosis and management. Note: HbA1c: glycated hemoglobin; DCCT: Diabetes Control and Complications Trial; FPG: fasting plasma glucose; NGSP: National Glycohemoglobin Standardization Program; OGTT: oral glucose tolerance test; T2DM: type 2 diabetes mellitus; ASCVD: atherosclerotic cardiovascular disease; CV: cardiovascular; HF: heart failure; CKD: chronic kidney disease; RASi: renin–angiotensin axis inhibitors; GLP-1 RA: glucagon-like peptide receptor agonist; SGLT2i: sodium–glucose cotransporter 2 inhibitors; eGFR: estimated glomerular filtration rate; DPP-4i: dipeptidyl peptidase 4 inhibitor [24,25]. Created with BioRender.com.

## 3. Renal Involvement in Patients with Diabetes Mellitus: What Not to Overlook

Chronic kidney disease (CKD) is currently an important epidemiological emergency [26]. The prevalence and incidence of kidney diseases worldwide are progressively increasing, mainly due to worsening global lifestyles (diet, obesity, sedentary lifestyle), but especially due to the increasing prevalence worldwide of the two main causes of CKD: arterial hypertension and diabetes mellitus (Figure 2).

Recently published data have shown that these are the two main causes of nephropathy worldwide, with diabetic nephropathy, in particular, being on the rise [8]. A recent investigation conducted by Stel et al. revealed that diabetic kidney disease (DKD) stands as the primary etiology of end-stage renal disease (ESRD) globally, detected in approximately 24–55% of patients undergoing renal replacement therapy. Specifically, this inquiry scrutinized the trajectory of diabetes prevalence as the principal cause of CKD on a global scale. Notably, it substantiated that the United States exhibits the highest incidence of patients undergoing replacement therapy owing to diabetic pathology. Nonetheless, it merits acknowledgment that despite the escalating incidence of ESRD attributable to diabetes in recent years, Europe demonstrates the smallest cohort initiating dialysis for diabetes, with an incidence spectrum that has progressively dwindled over time. This trend is likely attributed, in part, to a heightened emphasis on early detection and management of diabetic conditions alongside the increasingly targeted administration of more efficacious antidiabetic agents. Furthermore, it is important to recognize that renal involvement secondary to diabetes is likely still vastly underestimated [27]. In a recent study published in 2017 by O’Shaughnessy et al., histological alterations compatible with diabetic nephropathy were increasingly found in subjects undergoing renal biopsy [28].

In a recently published study, histological assessments conducted on autopsies of deceased individuals unveiled the prevalence of diabetic nephropathy in a substantial portion of patients who had not previously received a diagnosis of this condition [29]. Similar to most chronic nephrological conditions, renal abnormalities associated with diabetes mellitus initiate during the early stages of the disease. Initial hyperfiltration, precipitated by hemodynamic shifts occurring within the glomerulus, characterized by efferent arteriole vasoconstriction and afferent arteriole vasodilation, serves as the cornerstone for the onset of diabetic nephropathy. This phenomenon subsequently elevates glomerular pressure, fostering progressive impairment of the filtration barrier, emergence of microalbuminuria, and gradual reduction in nephron mass, accompanied by alterations in the glomerular filtration rate. It is imperative to note that while this pathology predominantly manifests at the glomerular level, it also subjects the tubules and endothelial cells of the kidney to stress, thereby culminating in comprehensive renal injury [30,31]. These pathophysiological mechanisms are described in Figure 3.

The aforementioned description pertains to and influences the progression of diabetic kidney disease (DKD) in its classical manifestation (classic phenotype), characterized by the presence of diabetic albuminuria, a strong correlation with glycated hemoglobin levels, and, often, the concurrent existence of diabetic retinopathy. The renal histopathological correlation is classified into four stages based on glomerular involvement. Class I, glomerular basement membrane thickening: isolated glomerular basement membrane thickening and only mild, nonspecific changes by light microscopy that do not meet the criteria of classes II through IV. Class II, mesangial expansion, mild (IIa) or severe (IIb): glomeruli classified as mild or severe mesangial expansion but without nodular sclerosis (Kimmelstiel–Wilson lesions) or global glomerulosclerosis in more than 50% of glomeruli. Class III, nodular sclerosis (Kimmelstiel–Wilson lesions): at least one glomerulus with a nodular increase in the mesangial matrix (Kimmelstiel–Wilson) without changes described in class IV. Class IV, advanced diabetic glomerulosclerosis: more than 50% global glomerulosclerosis with other clinical or pathologic evidence that sclerosis is attributable to diabetic nephropathy [33]. Recently, an alternative phenotype of DKD has emerged: the non-albuminuric variant. In this scenario, patients exhibit a decline in glomerular filtration rate (GFR) with a frequent absence of albuminuria. Notably, there is no association with diabetic retinopathy or glycated hemoglobin levels in these cases. This variant of DKD primarily stems from extraglomerular alterations, notably tubular dysfunction. Consequently, there are disruptions in tubular regeneration attributed to hyperglycemia and an increased susceptibility to tubular ischemia, partly influenced by the utilization of therapies such as renin–angiotensin system inhibitors (RASi) or sodium–glucose cotransporter 2 (SGLT2) inhibitors [34].

## 4. How to Reduce the Progression of Kidney Disease in Patients with DKD?

There is increasing interest in the use of therapeutic measures and lifestyle changes aimed at reducing the progression of kidney disease in patients with DKD. Last year, the recent KDIGO guidelines were published, which specifically addressed this issue [35]. In particular, the importance of acting on four critical points was emphasized: lifestyle, arterial hypertension, glycemic control, and nutrition (Figure 4).

It is important to remember that multiple interventions must be implemented in patients with DKD to achieve adequate control over a different group of clinical targets [35]. The management of hypertension holds paramount importance. Commencing treatment with renin–angiotensin system inhibitors (RASi) should be prioritized as the initial therapeutic approach, even in diabetic patients. Numerous studies, despite being somewhat dated, have evidenced their potential advantages in mitigating the advancement of chronic kidney disease (CKD) among individuals with diabetes mellitus. Vigilant monitoring within a span of 2–4 weeks post-initiation of therapy is imperative to assess any adverse impacts on renal function and potassium levels [36]. In this context, the integration of renin–angiotensin system inhibitors (RASi), which have previously demonstrated a pivotal role in managing diabetic kidney disease (DKD), with drugs possessing more targeted antidiabetic properties, such as SGLT2 inhibitors and glucagon-like peptide 1 (GLP1) antagonists, have been observed. This combination therapy, facilitating simultaneous intervention on various mechanisms, enables regulation of glucose metabolism, hemodynamics, and concomitant inflammation and fibrosis. Consequently, it yields highly favorable effects in attenuating the progression of renal damage [37].

Nutritional considerations also hold significant importance. Regrettably, there is a scarcity of randomized controlled trials investigating the impact of dietary interventions in patients with DKD, with most studies having a limited participant pool or focusing on short-term outcomes. Additionally, dietary recommendations vary in their level of restrictiveness based on the stage of CKD. For instance, managing potassium intake necessitates meticulous attention in the later stages of CKD, where potassium restriction is imperative, prompting patients to opt for foods (such as fruits and vegetables) with lower potassium content. Moreover, the incorporation of fruits and vegetables into the diet should align with guidelines pertinent to the primary diabetic condition [38].

The discourse on low-protein dietary regimens holds notable significance. For individuals with DKD not undergoing dialysis, maintaining a protein intake of 0.8 g/kg/day is recommended. However, restricting protein intake in these patients, who often receive advice to limit carbohydrates, fats, and alcohol, carries potential risks and could substantially diminish the caloric content of their diet. Conversely, for patients undergoing dialysis, a protein intake ranging between 1 and 1.2 g/kg/day is advocated. Sodium consumption assumes particular importance, with guidelines stipulating a daily intake of less than 2 g for individuals with diabetes and CKD. This recommendation stems from robust evidence indicating that reducing sodium intake leads to lowered blood pressure levels, with additional evidence of moderate quality suggesting favorable effects on cardiovascular diseases, stroke risk, and the progression of CKD [39].

Optimal glycemic management stands as the principal determinant in attenuating the progression of chronic kidney disease (CKD) among diabetic individuals. Current guidelines for glucose regulation, as indicated by glycated hemoglobin levels, underscore the importance of adaptable targets, acknowledging the necessity of discerning individualized factors to establish optimal treatment strategies. Monitoring this parameter should occur minimally twice annually, with a possibility of up to four times for patients encountering challenges in glycemic control. Furthermore, it is vital to recognize a significant consideration: glycated hemoglobin measurement may possess notable limitations in individuals with CKD, particularly those in advanced stages. This limitation stems from the potential manifestation of heightened oxidative stress, inflammation, and metabolic acidosis, factors that can instigate the formation of advanced glycation end products, consequently elevating glycated hemoglobin levels. Conversely, it is essential to acknowledge that individuals with advanced renal disease may exhibit lower levels of hemoglobin than anticipated, attributed to the prevalent anemia often observed in this population.

## 5. Kidney and Diabetes: Special Populations

Here, we will outline the distinct characteristics of two categories of nephropathic patients: those undergoing hemodialysis and those who have received kidney transplants. These distinctions are critical for their comprehensive evaluation and clinical management.

Patients undergoing hemodialysis, particularly those with a lengthy history of diabetes mellitus, exhibit specific features. They undergo extracorporeal therapy thrice weekly. Neurological system alterations such as polyneuropathy and diminished baroreceptor sensitivity, along with cardiovascular system modifications, are common in these individuals. Moreover, they often experience pronounced thirst, resulting in considerable weight gain before sessions and necessitating exposure to high ultrafiltration rates. Consequently, there exists a notable risk of intradialytic hemodynamic instability, a factor essential for consideration when devising dialysis schedules [39]. Furthermore, following the start of the dialysis session, an initial spontaneous correction of hyperglycemia frequently observed in these patients is defined as “Burnt-Out Diabetes”. Intradialytic hypoglycemia is something that is very frequent and must always be considered in these patients, modulating specific antidiabetic therapies before the hemodialysis session [40]. Another unique circumstance involves diabetic kidney transplant recipients. These individuals may exhibit pre-existing diabetes prior to transplantation, or alternatively, they may develop diabetes subsequent to the transplant procedure. Literature reports indicate an incidence of post-kidney transplant diabetes mellitus ranging from 5 to 25% among recipients [41]. Immunosuppressive drugs play an important role in this, practically all of which can determine alterations in insulin production and reduced insulin sensitivity [42]. This complicates the clinical oversight of diabetes in these patients significantly. Special consideration must be given to states of glucose metabolism disruption that have not progressed to overt diabetes. Such conditions may elevate the risk of developing diabetes shortly or long after transplantation and also heighten the susceptibility to significant clinical complications. Consequently, it is advised to conduct an oral glucose tolerance test (OGTT) six months post-transplantation in non-diabetic kidney transplant recipients or, in certain instances, prior to listing for transplantation.

## 6. CKD Not Related to Diabetes

In recent years, there has been a growing inclination towards recommending renal biopsy for patients diagnosed with CKD and diabetes mellitus. Increasing evidence in the literature highlights the existence of nephropathies distinct from diabetic nephropathy in individuals with diabetes mellitus (NDKD). Special attention is warranted toward recognizing signs and symptoms that may raise suspicion of NDKD. A study published approximately four years ago retrospectively examined renal biopsies conducted on 832 patients with diabetes mellitus between 2002 and 2014. Predominantly, the indication for biopsy in these patients was the presence of nephrotic syndrome or a nonspecific elevation in proteinuria. Notably, nearly two-thirds of patients who underwent biopsy exhibited NDKD as the primary etiology of their kidney disease. Multivariate analysis identified the presence of microhematuria and the absence of diabetic retinopathy as the variables most predictive of diagnosis [43]. Recently, Fiorentino et al. conducted a meta-analysis encompassing over 48 studies, revealing highly variable prevalence rates for diabetic kidney disease (DKD) ranging from 6.5% to 94% of histological diagnoses and for non-diabetic renal disease (NDRD) ranging from 3% to 82% of cases. Particularly noteworthy was the discovery of mixed forms, albeit their prevalence varied significantly, accounting for 4% to 45% of cases. IgA nephropathy emerged as the most prevalent cause of NDRD [44]. The conclusions of this study are that renal biopsy can be fundamental in clarifying the epidemiology of kidney disease in patients with diabetes and in planning, where necessary, specific treatment.

## 7. Hepatic Involvement in Patients with Diabetes Mellitus: What Not to Overlook

Non-alcoholic fatty liver disease (NAFLD), the most common liver disease in Western countries, is characterized by the presence of hepatic fat in the absence of other causes of liver disease, and its prevalence is expected to dramatically increase in the next years, mirroring the spread of metabolic alterations and unhealthy lifestyle [45]. NAFLD encompasses a wide range of liver diseases, from simple steatosis to steatohepatitis, where inflammation coexists, to hepatic fibrosis and cirrhosis. NAFLD is strictly related to metabolic alterations. Therefore, in June 2023, its nomenclature was changed to metabolic dysfunction-associated steatotic liver disease (MASLD), defined by the presence of hepatic fat and at least one metabolic alteration [46]. Along with metabolic alterations, MASLD also recognizes in its pathogenesis a genetic predisposition mainly driven by three polymorphisms, the strongest one being mutations in the Patatin-like phospholipase domain protein 3 (PNPLA3) gene.

Among all metabolic dysfunctions, T2DM is surely one of the most closely linked to MASLD. In fact, the prevalence of MASLD in the general population is about 25–30%, but it reaches up to 60% in diabetics [47] (Figure 5). Indeed, MASLD and T2DM share several pathogenetic risk factors, such as an unhealthy lifestyle and obesity, and expose patients to increased morbidity and mortality, mainly driven by cardiovascular disease.

In fact, both MASLD and T2DM determine atherogenic dyslipidemia, hepatic and/or systemic insulin resistance, hyperglycemia, dysbiosis, increased secretion of pro-inflammatory biomarkers (i.e., CRP, IL-6, TNF), and a pro-coagulant profile [48] (Figure 6).

**Figure 5 ijms-25-07728-f005:**
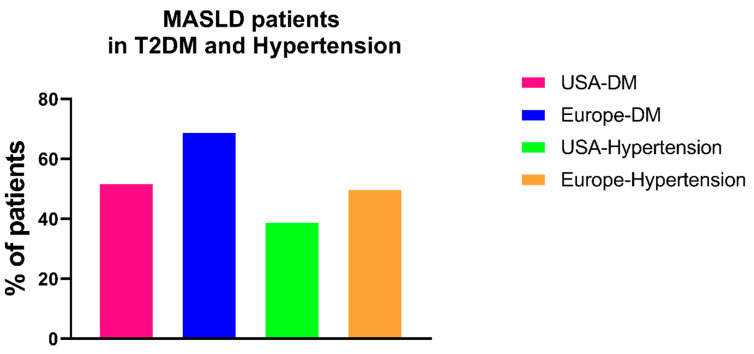
Main causes of MASLD in Europe and the USA. Note: Prevalence of MASLD in the patients with T2DM and hypertension in the USA and Europe [45,49].

**Figure 6 ijms-25-07728-f006:**
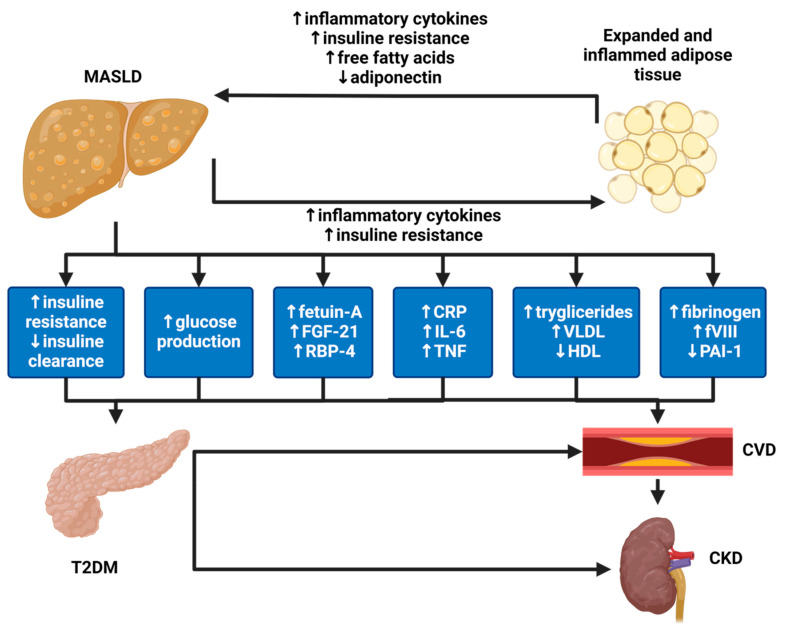
Link between MASLD, T2DM, and CKD. Note: MASLD augments the risk profile for atherothrombotic events through several mechanisms leading to CVD and, subsequently, CKD development. These include the promotion of atherogenic dyslipidemia characterized by elevated triglyceride levels, reduced HDL cholesterol levels, and heightened levels of small, dense LDL particles. Additionally, MASLD contributes to hepatic and/or systemic insulin resistance, dysglycemia, and increased secretion of various pro-inflammatory biomarkers (e.g., CRP, IL-6, TNF) and hemostatic-fibrinolytic factors (e.g., fibrinogen, factor VII, von Willebrand factor, and PAI-1). Moreover, MASLD predisposes individuals to the future onset of T2DM by enhancing glucose production in the liver and exacerbating hepatic and systemic insulin resistance. This may occur through the systemic release of pro-inflammatory factors and certain liver-secreted proteins possessing diabetogenic properties, such as fetuin-A, FGF-21, and RBP-4. Furthermore, T2DM is a recognized risk factor for the progression of MASLD, advancing from simple steatosis to non-alcoholic steatohepatitis NASH, cirrhosis, and, in some instances, hepatocellular carcinoma. T2DM, on the other hand, is a key mediator of CKD and CVD development. CRP: C-reactive protein; CVD: cardiovascular disease; FGF-21: fibroblast growth factor 21; PAI-1: plasminogen activator inhibitor 1; RBP-4: retinol-binding protein 4; T2DM: type 2 diabetes mellitus; TNF: tumor necrosis factor [50]. Created with BioRender.com.

Finally, it is important to stress how MASLD and T2DM mutually negatively impact their clinical course. In fact, if, on the one hand, T2DM is a known risk factor for the onset of MASLD and its progression to non-alcoholic steatohepatitis (MASH) and fibrosis, in turn, MASLD predisposes to T2DM development and impairs its glycemic control [51,52].

A study conducted in a small cohort of Finnish subjects showed that diabetic patients had higher intrahepatic triglycerides as assessed by magnetic resonance compared to matched BMI and waist circumference controls, and similarly, they had 20 to 400% increased hepatic fat compared to non-diabetic individuals at same ALT levels [53]. Another wide cohort study by Wang et al. following up more than 30,600 subjects for approximately 3 years, showed that T2DM exposed patients to a nearly 60% risk of developing MASLD over time independently of BMI, lifestyle, age, or sex, and the higher the glycemic impairment (i.e., pre-diabetes vs. established diabetes), the higher the risk [54]. Similarly, T2DM has been shown to be an independent risk factor also for the worsening of MASLD over time, with a 70% risk of progressive fibrosis in 12 years [55]. Notably, T2DM seems to cooperate with the genetic predisposition to the development of advanced forms of MASLD, and preventing or reversing T2DM and obesity in persons carrying the mutated variant of PNPLA3 may lower the risk of progressive liver disease [56].

On the other hand, the same study by Wang et al. showed that the incidence of pre-diabetes and T2DM were higher in the NAFLD groups compared to the non-NAFLD one, with a hazard risk of about 30% and 90%, respectively [54]. This evidence was confirmed by a meta-analysis by Mantovani et al., which included 33 observational studies with more than 500,000 individuals (30.8% with MASLD) and with 27,953 cases of incident diabetes over a median of 5 years. The authors demonstrated that patients with MASLD had a higher risk of incident diabetes than those without MASLD (random-effects HR 2.19), and the more severe the liver disease, the higher the risk (random-effects HR 3.42 for hepatic fibrosis) [57]. Furthermore, another study following up more than 4000 patients (16% with MASLD) demonstrated not only that those with baseline steatosis had a 2.5-fold increased risk of developing T2DM over time but also that in those with resolved hepatic steatosis, the incidence of T2DM was markedly reduced compared to those who remained steatotic at follow-up (6% vs. 18% *p* < 0.001) [58]. Finally, the presence of MASLD, particularly if fibrosis coexists, is associated with an increased risk of vascular diabetic complications [59,60,61].

### Screening Diabetic Patients for MASLD

Given the close mutual relationship between T2DM and MASLD, international guidelines from both hepatology and diabetes societies strongly recommend screening the diabetic population for liver disease [62].

Currently, several alternatives are available for the assessment of liver disease, either invasive or non-invasive. Hepatic steatosis is currently easily detectable and graded by liver ultrasound, which is recommended as first-line screening by international guidelines, despite low sensitivity for lower grades of hepatic fat (i.e., <20%). However, magnetic resonance imaging-PDFF, which is the gold standard, is too expensive and not applicable to the wide cohorts of the diabetic MASLD population, so its use is discouraged in clinical settings. Some steatosis scores have also been developed, mainly the fatty liver index (combining in its formula triglycerides, GGT, BMI, and waist circumference, being positive if >60) and the hepatic steatosis index (combining in its formula AST, ALT, BMI, and T2DM, being positive if >36) and can be used whenever imaging tools are not available or for wide epidemiologic studies [49,63].

For liver fibrosis, liver biopsy remains the gold standard for the staging of MASLD, even though it is limited by the cost and invasiveness of the procedure. Therefore, over the last few years, non-invasive methods have accumulated, especially transient elastography (Fibroscan) and fibrosis scores, combining clinical and biochemical data. The most used is the fibrosis-4 score (FIB-4) (combining in its formula AST, ALT, age, and platelets, being positive if >2.67 and negative <1.3). Fibroscan combines ultrasound techniques with the propagation of a mechanical wave into the liver, assessing the stiffness of the parenchyma as a surrogate of fibrosis by providing a parameter called liver stiffness measurement (LSM). Fibroscan also gives another variable called the controlled attenuation parameter (CAP), which is an indirect measurement of hepatic fat [64]. The American Association for the Study of the Liver has recently proposed an algorithm for the screening and referral of patients with suspected liver disease by combining FIB-4 and Fibroscan, which can also be applied to the diabetic population (Figure 7) [63].

Even though the recommendation from the guidelines is the large-scale screening of diabetic patients, data in the literature help us identify those at higher risk of advanced liver disease and who may benefit the most from a referral.

Among T2DM subjects, those with overweight/obesity and particularly those with increased visceral adiposity, represented by deranged waist circumference, are more likely to have hepatic fibrosis [60,61,65]. Similarly, alterations in the lipid profile, especially hypertriglyceridemia, insulin resistance, and glycemic control, are risk factors for hepatic fibrosis [66,67,68,69].

Notably, all these studies demonstrated that transaminases are independently associated with the presence of hepatic fibrosis despite remaining within normal ranges. This evidence is widely reported in the literature, as described by Fracanzani et al. in a cohort of biopsy-proven MASLD patients, where the severity of liver damage was independent of deranged transaminases, and this was even more pronounced in the diabetic population [70]. In addition, another small study involving 54 diabetic subjects and 26 healthy controls, all with normal transaminases, revealed a significantly higher prevalence of hepatic steatosis (46% vs. 12%) by ultrasound and fibrosis by Fibroscan (12% vs. 0%) in the diabetic population [71]. Finally, Lombardi et al. followed up a cohort of diabetic patients for 5 years, registering a worsening of fibrosis in approximately 40% of the cohort in the presence of normal transaminases both at baseline and follow-up in almost the totality of the cohort [69]. Therefore, all this evidence possibly speculates on the need for new thresholds of transaminases in the diabetic population.

## 8. How to Reduce the Progression of MASLD in Patients with T2DM?

Therapies to treat T2DM are accumulating and becoming promising; however, currently, only Resmetirom, an agonist of the thyroid hormone receptors, has been approved by the Food and Drug Administration for the treatment of MASLD. However, it is not widely available worldwide [72]. Therefore, currently, modifications in lifestyle, especially diet and regular physical exercise, remain the standard of care in the treatment of this condition [63].

Indeed, weight loss, either achieved with diet or physical exercise or their combination, has been demonstrated to promote improvement in hepatic steatosis and fibrosis, especially if the loss is >5–10% from baselines [73,74,75]. Similarly, a randomized clinical trial on 300 diabetic patients demonstrated a complete resolution of the diabetic disease in the arm of treatment with diet, achieving weight loss compared to the arm addressed with standard of care, and the greater the weight loss, the higher the percentage of diabetic resolution, up to 87% in the group with >15% of weight loss from baseline [76].

Many dietary approaches have been proposed; however, the Mediterranean diet (MD) has been shown to improve MASLD [77] as well as to reduce cardiovascular events [78]. Similarly, the MD has been shown to improve insulin resistance, with some benefits also in patients with T2DM [79].

The MD is characterized by reduced daily intake of carbohydrates (30% of the whole daily calorie intake, ideally whole grain) and high fat consumption (35–45% of the total energy intake), mainly mono-unsaturated fatty acids and poly-unsaturated fatty acids (contained in olive oil, nuts, and fish) and reduced saturated fatty acids. Sugary drinks and high fructose corn syrup should be avoided, whereas fibers found in vegetables, legumes, and whole grains are encouraged [77]. A small amount of alcohol consumption, below 30 g/day in men and 20 g/day in women, is permitted [64].

As for physical activity, patients with MASLD are stimulated to perform at least 150 min/week of moderate-intensity physical activity over 3–5 sessions, combining both aerobic and resistance training [80]. This evidence is also supported by a recent study involving 233,676 men and women followed up for 5 years. Moderate to vigorous exercise (≥5 times per week) was a protective factor for incident fatty liver (HR 0.86; 95% CI 0.80,0.92) and a favoring factor for its resolution (HR 1.40; 95% CI 1.25,1.55) [81].

Glycemic control prevents the progression of MASLD in patients with T2DM. In fact, an increase in HbA1c is associated with a higher risk of having advanced forms of liver disease, either as higher grades of steatosis or severity of histological fibrosis [82]. In particular, in a study involving more than 700 patients with biopsy-proven MASLD, every 1% increase in mean HbA1c was associated with 15% higher odds of worsening in the fibrosis stage [83] and more unstable glycemic control in the past years before liver biopsy, the higher the risk having steatohepatitis and advanced fibrosis.

Finally, antidiabetic drugs are currently under investigation for the treatment of MASLD in several clinical trials. Only Resmetirom, an agonist of the thyroid hormone receptors, has been recently approved by the Food and Drug Administration for the treatment of MASLD; however, it is not widely available worldwide [72]. In addition, also the use of pioglitazone (a member of thiazolidinediones) is recommended by the EASL-EASD-EASO and AASLD practice guidelines for the treatment of non-cirrhotic adults with biopsy-confirmed MASH [63,64] even though its use is still off label in MASLD patients and its affected by many side effects as the predisposition to cardiovascular events, weight gain, fluid retention, and risk of bone fractures. Therefore, several other drugs have been tested in randomized clinical trials as potential therapies for MASLD; among them, glucagon-like peptide-1 receptor agonists (GLP-1RAs) (liraglutide or semaglutide) have attracted interest for their ability to promote weight loss, improving glycemic control and insulin resistance [84], as well as reducing liver fat and resolving NASH without worsening liver fibrosis [85,86]. Another molecule that seems promising in the treatment of MASLD is the sodium–glucose cotransporter-2 inhibitor (SGLT2i), which has been shown to reduce serum transaminase levels and improve hepatic fat content [87], with beneficial effects on liver fibrosis reported [88,89,90].

## 9. Liver and Diabetes: Special Populations

Among MASLD patients, two special populations who should be treated separately are those with hepatic cirrhosis, possibly evolving to end-stage liver disease, and those with hepatocellular carcinoma (HCC).

As previously said, MASLD is a progressive disease that may evolve in some cases to hepatic cirrhosis, which, in turn, can remain stable over time or decompensate up to end-stage liver disease, exposing patients to death if a liver transplant is not provided. In particular, some of the typical clinical features of hepatic decompensation are variceal or gastrointestinal bleeding, ascites, and hepatic encephalopathy. A recent meta-analysis including six international centers with a total of 2016 participants (736 with T2DM) pointed to diabetes as a key determinant of hepatic decompensation in MASLD subjects, independently of BMI, race, and baseline liver stiffness [91]. In addition, a recent study applying a polygenic risk score for the prediction of incident cirrhosis, decompensated liver disease, hepatocellular carcinoma, and liver transplantation during a median follow-up of 9 years in a cohort of 266,687 individuals of the UK Biobank highlighted how the presence of T2DM further increased the risk of these adverse outcomes of about 2.6 to 5.7-fold in patients with an unfavorable genetic and fibrosis profile at baseline [92].

Hepatocellular carcinoma (HCC) is the sixth leading cause of cancer worldwide, being related to high morbidity and mortality and impairment in the quality of life of patients. In the past years, viral hepatitis was the most frequent cause of progressive liver disease up to HCC, but in the last decades, MASLD has become the leading etiology of this condition, as well as the leading cause of HCC-related transplant in Western countries [93].

HCC typically arises in a cirrhotic liver, although in some cases, it can occur in a non-cirrhotic one, with some grades of fibrosis and inflammation present, and very seldom in a liver characterized only by fat accumulation [94]. Fibrosis, either assessed by histology or Fibroscan, is a risk factor for HCC but also metabolic comorbidities, especially T2DM and obesity [95,96]. In particular, a prospective study enrolling 17,000 subjects over a 30-year follow-up has shown that the occurrence of HCC in 119 subjects was independently related to the presence of baseline T2DM with a 5-fold increased risk, especially for longer the duration of T2DM or coexistence with other metabolic alterations [97]. Finally, T2DM is related also poorer overall survival and poorer disease-free survival in HCC patients [98].

### The Kidney and the Liver Disease: Two Faces of the Same Disease

Given the fact that T2DM is a risk factor for the onset of both kidney and liver disease, an association between these two conditions is expected. In particular, original data and meta-analysis support the independent association of MASLD with an increased risk of prevalent and incident CKD, and the more severe the liver disease, the higher the risk [99,100]. The pathophysiology of this evidence relies on common pathogenetic mechanisms and pathways such as systemic insulin resistance, atherogenic dyslipidemia, presence of hypertension, and activation of the renin-angiontensin system, as well as endothelial dysfunction and a pro-inflammatory and pro-coagulant profile [101].

In addition, the coexistence of CKD and MASLD in T2DM implies more targeted management concerning lifestyle and antidiabetic therapies. In particular, as said, patients with CKD are encouraged to limit the amount of total protein intake to 0.8 g/kg/day. However, patients with both CKD and MASLD are advised to also restrict carbohydrates and fats, possibly diminishing the caloric content of their diet and limiting their food choices. Possibly, suggesting a higher consumption of vegetables, legumes, and whole grains could improve this. As for therapy, it is clearly evident that in patients with CKD and MASLD, the introduction of SGLT2i could be the first choice for the beneficial effects demonstrated on both organs, and when contraindicated, the use of GLP-1 agonists could be a valid alternative for its positive impact, as well. Finally, renin–angiotensin system inhibitors (RASi) have a pivotal role in the management of DKD, and some data report a positive effect in preventing fibrosis progression in MASLD as they seem to reduce oxidative stress and block activation of hepatic stellate cells [102]. For sure, obtaining glycemic control prevents the progression of both CKD and MASLD in patients with T2DM, whatever the strategy to achieve an optimal HbA1c.

## 10. Conclusions

In conclusion, type 2 diabetes (T2DM) is a multifaceted and systemic disease that may affect different organs and systems beyond the classical vascular involvement. In this review the authors have explored some important and sometimes unexplored aspects of both renal and hepatic implications. Indeed, being highly progressive, T2DM may easily cause deterioration of affected organs with dramatic patient morbidity and mortality, often silently and asymptomatically until the irreversible end-stage phase of the disease. For this reason, it is important not to forget very simple assessments in diabetic patients in order to detect and treat promptly liver and CKD and prevent the occurrence of advanced complications (Table 1).

As for the kidney, we must remember that DKD still represents the main cause of end-stage renal disease worldwide. The therapeutic options available today make early and timely nephrological evaluation mandatory in a multi-specialist follow-up context. The presence of renal pathologies different from diabetic ones, even in diabetic subjects, keeps open the question regarding at least the indications for renal biopsy in these patients. Certainly, renal biopsy, in selected cases, could allow the identification and treatment of cases where CKD is not fueled by diabetic pathology but by other types of nephropathies.

As for the liver, diabetes represents the key determinant of the onset and progression of MASLD to advanced forms up to carcinogenesis and end-stage liver disease. In turn, hepatic steatosis promotes difficult glycemic control and the onset of microvascular complications, including those determining DKD. Therefore, screening the diabetic population for liver disease becomes mandatory, especially in the presence of other metabolic alterations and irrespective of transaminases, also given the availability of non-invasive tools, such as the FIB-4 score and Fibroscan. In addition, new antidiabetic drugs, such as GLP-1ARs or SGLT2i, may open up a new scenario in the treatment of liver disease in this population, going beyond simple glycaemic control (Figure 8), which remains the cornerstone of MASLD treatment, together with a healthy lifestyle.

## Figures and Tables

**Figure 2 ijms-25-07728-f002:**
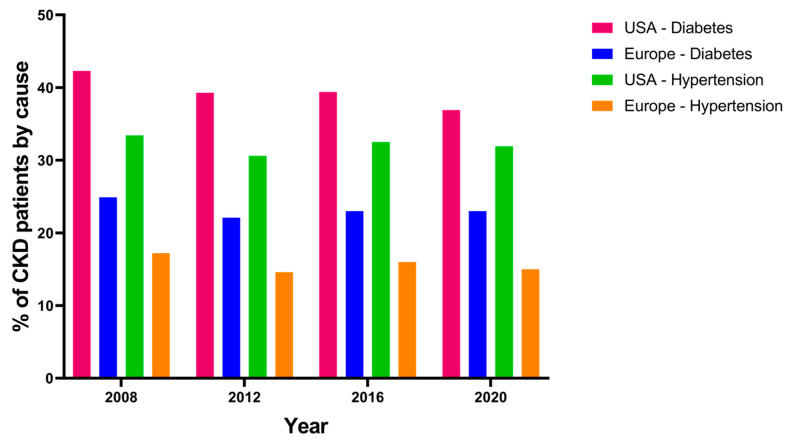
Main causes of CKD in USA and Europe in the last 20 years. Note: Data obtained from USRDS (https://usrds-adr.niddk.nih.gov/ accessed on 29 June 2024) and ERA-Registry (https://www.era-online.org/research-education/era-registry/annual-reports/ accessed on 29 June 2024). CKD: chronic kidney disease.

**Figure 3 ijms-25-07728-f003:**
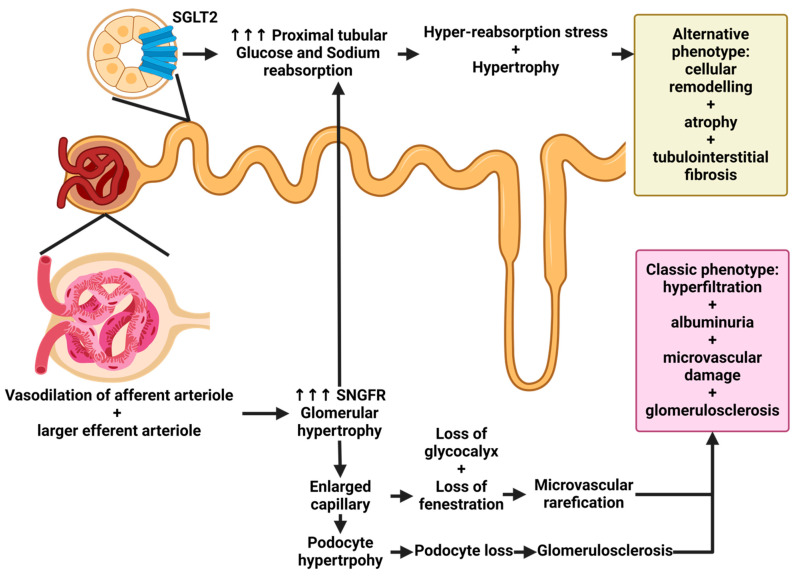
Overview of the pathophysiology of diabetic kidney disease. Note: The typical kidney comprises autonomous functional units termed nephrons. The dimensions of the filtration surface and the barrier across the glomerular capillaries are tailored to the physiological demands of filtration load, yielding an average total glomerular filtration rate (GFR) of approximately 120 mL/min. The quotient of total GFR divided by the number of nephrons delineates the single-nephron GFR (SNGFR). Most filtered solutes, including glucose and NaCl, undergo reabsorption primarily in the proximal tubule. In individuals with diabetes mellitus (DM), elevated blood glucose levels prompt augmented glucose reabsorption via the sodium–glucose cotransporter 2 (SGLT2) located in the proximal tubule. This heightened glucose reabsorption elicits lysosomal stress within proximal tubule cells, diminishes sodium delivery to the macula densa, deactivates tubuloglomerular feedback, and induces dilation of the afferent arteriole, culminating in sustained glomerular hyperfiltration. Concurrent activation of the renin–angiotensin system exacerbates SNGFR and glomerular hypertension. Glomerular hypertension instigates a transforming growth factor-α (TGFα)-mediated augmentation in the filtration surface of the glomerular filtration barrier (GFB). Additionally, increased SNGFR precipitates elongation of the proximal tubule, contributing to the observed renomegaly in diabetic kidney disease (DKD). Obesity, a prevalent precipitant of type 2 DM (T2DM), exacerbates the aforementioned effects due to expanded body fluids and filtration load. Pre-existing chronic kidney disease (CKD) involves hypertrophy and loss of remnant nephrons, irrespective of diabetic effects. Nevertheless, newly developed DM can intensify SNGFR and aggravate glomerular hypertension, thereby prompting further expansion of the glomerular filtration surface. Excessive podocyte hypertrophy beyond a critical threshold leads to podocyte detachment, macroproteinuria, glomerulosclerosis, and nephron loss. The confluence of CKD and DM accelerates vascular aging and endothelial and tubular dysfunction, ultimately precipitating renal ischemia and hastened CKD progression. SGLT2: sodium–glucose transport protein 2; SNGFR: single nephron glomerular filtration rate [32]. Created with BioRender.com.

**Figure 4 ijms-25-07728-f004:**
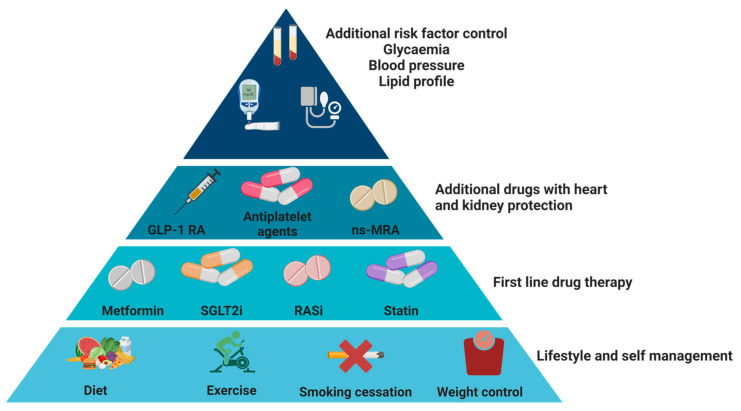
Summary of the recent indications for diabetes and CKD management. Note: Individuals afflicted with diabetes and chronic kidney disease (CKD) necessitate a holistic therapeutic regimen aimed at ameliorating both renal and cardiovascular outcomes. This regimen should encompass lifestyle modifications and self-administration practices as foundational strategies for all patients. Additionally, it should entail the implementation of primary pharmacological treatments tailored to individual clinical profiles, with subsequent layers of adjunctive medications proven to confer renal and cardiac protection based on evaluations of residual risk. Furthermore, supplementary interventions should be employed as required to further mitigate risk factors. For glycemic management, individuals with type 1 diabetes (T1D) should primarily rely on insulin therapy, while those with type 2 diabetes (T2D) should utilize a combination of metformin and sodium–glucose cotransporter-2 inhibitors (SGLT2i). Metformin may be administered when the estimated glomerular filtration rate (eGFR) is ≥30 mL/min per 1.73 m^2^, and initiation of SGLT2i is recommended when eGFR is ≥20 mL/min per 1.73 m^2^, continuing until dialysis or transplantation becomes necessary. Renin–angiotensin system (RAS) inhibition is advised for individuals exhibiting both albuminuria and hypertension (HTN). Moreover, all patients with T1D or T2D and CKD are recommended to receive statin therapy. In cases where glycemic targets are not achieved with standard therapies or if patients are intolerant to them, glucagon-like peptide-1 receptor agonists (GLP-1 RA) are the preferred alternative for individuals with T2D. Additionally, for patients with T2D and elevated residual risks of kidney disease progression and cardiovascular events, as evidenced by persistent albuminuria (>30 mg/g [>3 mg/mmol]), nonsteroidal mineralocorticoid receptor antagonists (ns-MRAs) may be added to the primary therapeutic regimen. As for antiplatelet therapy, lifelong administration of aspirin is recommended for secondary prevention among those with established cardiovascular disease. Furthermore, consideration may be given to its use for primary prevention among patients deemed to have a high risk of atherosclerotic cardiovascular disease (ASCVD) [35].

**Figure 7 ijms-25-07728-f007:**
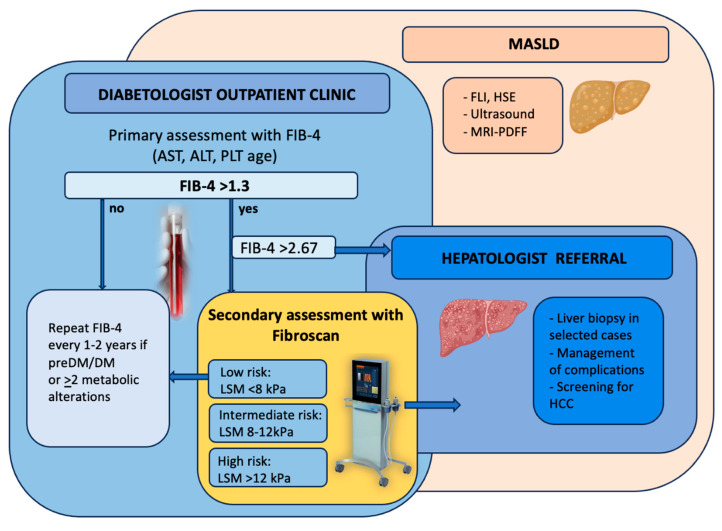
Proposed algorithm for the screening of MASLD and fibrosis in patients with T2DM. Note: Patients with type 2 diabetes should be screened for MASLD independently of transaminases by either liver ultrasound, non-invasive tests (FLI, HSE), or magnetic resonance (MRI-PDFF). Once hepatic steatosis is detected, hepatic fibrosis should be ruled in or out using non-invasive scores, mainly the FIB-4. If the Fib-4 is <1.3, the presence of fibrosis is unlikely, and the patient should be re-evaluated every 1–2 years. If the FIB-4 is >2.67, the presence of fibrosis is likely, and the patient should be referred to a hepatology center. In case of indeterminate FIB-4 values (>1.3 but <2.67), the patient should undergo a Fibroscan to further stratify the risk of fibrosis and the indication to a hepatological referral. FLI: fatty liver index; HSI: hepatic steatosis index; MRI-PDFF: magnetic resonance imaging-PDFF; FIB-4: fibrosis-4 score; LSM: liver stiffness measurement; HCC: hepatocellular carcinoma; DM: type 2 diabetes.

**Figure 8 ijms-25-07728-f008:**
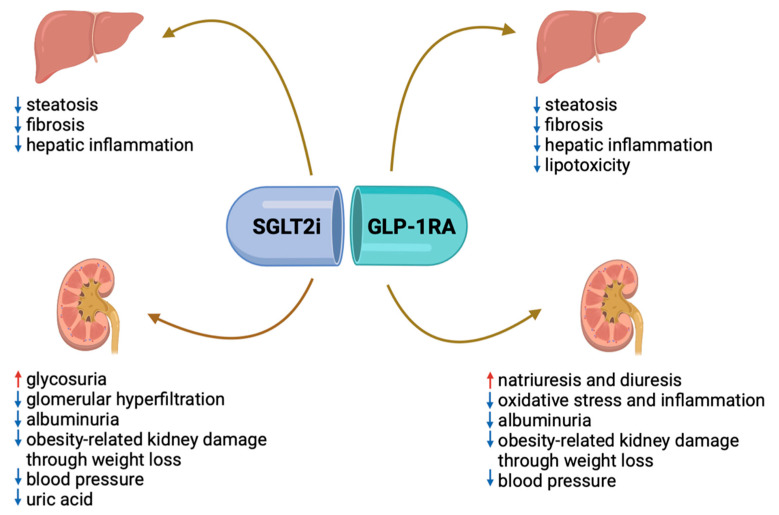
Mechanisms of action of SGLT2i and GLP-1ARs on kidney and liver. **Note:** The GLP1-RA acts both on the liver and the kidney. In the former, they favor a decrease in hepatic inflammation and lipotoxicity, with a consequent decrease in steatosis and fibrosis; in the latter, they favor an increase in natriuresis with consequent increased diuresis and decreased blood pressure and albuminuria. In addition, by favoring weight loss, GLP1-RA prevents obesity-related kidney damage mediated by inflammation and adipokines. SGLT2i acts on the liver and kidney, too. In the hepatic setting, they promote reduced inflammation and consequent reduced steatosis and fibrosis, also mediated by weight loss. In the kidney, SGLT2i fosters glycosuria with consequent reduced glomerular hyperfiltration and albuminuria and reduced kidney damage. In addition, by fostering glycosuria, they reduce blood pressure and hypertension-related kidney disease. Abbreviations: GLP-1 RA: glucagon-like peptide receptor agonist; SGLT2i: sodium–glucose cotransporter 2 inhibitors. Created with BioRender.com.

**Table 1 ijms-25-07728-t001:** Addressing indications for clinicians for a comprehensive evaluation of MASLD and CKD in patients with T2DM in order to manage alterations and avoid complications.

Assessment	Abnormal Cut-Offs	Endpoint	Complications to Avoid
Non-invasive scores - FLI (triglycerides, GGT, BMI, and waist circumference) - HSI (AST, ALT, BMI, and T2DM) - FIB-4 score (AST, ALT, PTL, age)	>60 >36 >2.67	Hepatic steatosis Hepatic steatosis Hepatic fibrosis	- Cirrhosis development with risk of hepatocellular carcinoma and hepatic decompensation, up to liver transplant - Worsening of vascular complications
Liver ultrasound	Bright pattern compared to the kidney	Hepatic steatosis	
Fibroscan >8 kPa	>8 kPa	Hepatic fibrosis	
GFR by creatinine clearance (Urinary creatinine (mg/dL) urine volume mL)/(Plasmatic creatinine mg/dL) or CKD-EPI formula	<60 mL/min	CKD	- End-stage CKD with the necessity of dialysis or kidney transplantation
Microalbuminuria	>30 mg/dL in the morning spot urine	CKD	
Glycated hemoglobin	>53 mmol/mol	T2DM control	- Vascular and hepatic complications and increased CV mortality

Note: In the presence of T2DM, clinicians should test patients for both hepatic and kidney disease. As for the liver, patients could be tested either by non-invasive scores of steatosis (FLI and HSI) and fibrosis (FIB-4) or by imaging such as liver ultrasound and Fibroscan for the detection of steatosis and fibrosis, respectively. As for the kidney, patients should be tested for GFR or by approved formula (the CKD-EPI 2023) [103] or by direct measurement of creatinine clearance, as well as for the presence of microalbuminuria in the morning urine. Patients should also be tested for glycated hemoglobin to check for diabetes control. These tests are all useful for the early detection of the presence of hepatic or renal disease and to promptly apply strategies to control these diseases and prevent their progression [104]. Abbreviations: FLI: fatty liver index; BMI: body mass index; HSI: hepatic steatosis index; T2DM: type 2 diabetes; FIB-4: fibrosis-4 score; PTL: platelets; GFR: glomerular filtration rate.

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
