# Peer review of "What Not to Overlook in the Management of Patients with Type 2 Diabetes Mellitus: The Nephrological and Hepatological Perspectives"

_ijms, 2024, doi:10.3390/ijms25147728_

Round 1

Reviewer 1 Report

Comments and Suggestions for Authors

This article revises diabetes mellitus, mainly focused on the effects of kidney and liver. The manuscript is well written, it is easy to read and understand.

 The manuscript may benefit from a description of more general aspects of the disease, including:

 (1) The pathophysiology of type 2 DM could be expanded including insulin resistance, impaired insulin secretion, genetic susceptibility, role of diet, obesity and inflammation, role of intrauterine development, drug-induced hyperglycemia (please pay attention to immune checkpoint inhibitors).

 (2) A more detailed description of the clinical presentation, diagnosis, and initial evaluation of diabetes mellitus in adults. Clinical presentation (type 2 and type 1 diabetes), diagnostic criteria (hyperglycemia, prediabetes, diagnostic tests), differential diagnosis (other causes of hyperglycemia), evaluation (biochemical testing).

 (3) Management/treatment of DM in a general manner.

 (4) The pathology of the lesions of DM could be added. Renal Pathology Society classification. Class I, II, III, and IV.

 (5) Line 137. Could you please add the website link of the KDIGO guidelines?

 (6) Histological images (if available) could be included.

 (7) A table with the different types of drugs recommended in the treatment may help the readers.

 (8) These articles provides nice figures.

 Mohandes S, Doke T, Hu H, Mukhi D, Dhillon P, Susztak K. Molecular pathways that drive diabetic kidney disease. J Clin Invest. 2023;133(4):e165654. Published 2023 Feb 15. doi:10.1172/JCI165654

 Friedman SL, Neuschwander-Tetri BA, Rinella M, Sanyal AJ. Mechanisms of NAFLD development and therapeutic strategies. Nat Med. 2018;24(7):908-922. doi:10.1038/s41591-018-0104-9

Author Response

We thank the reviewer for all his suggestions. We appreciate the insightful comments made by the reviewer. In response, we made our best efforts to improve the manuscript. Please see the attached file "Reviewer 1" for our replies. All corrections in the text are written in red and highlighted in yellow.

Reviewer 2 Report

Comments and Suggestions for Authors

The authors present a review article that aims to present the main characteristics and crucial points not to be overlooked regarding the renal and hepatic involvement in DM patients focusing on the inter-relationships between the renal and the hepatic involvements.

 I would like to raise the following concerns.

1.

Figure 1. Main Causes of CKD Prevalence: Arterial Hypertension and Diabetes Mellitus in the USA and Europe Over the Past 20 Years.

It is also suggested to provide information on the prevalence of MASLD or hepatic disorders: arterial hypertension and diabetes mellitus.

2.

In conclusion (lines 523-526): In addition, new antidiabetic drugs, such as GLP-1ARs or SGLT2i, may open up a new scenario in the treatment of liver disease in this population, going beyond simple glycaemic control, which remains the cornerstone of MASLD treatment, together with a correct lifestyle.

Sodium-glucose cotransporter 2 inhibitors (SGLT2is) are a potent class of antidiabetic drugs that act by inducing glycosuria through the inhibition of glucose reabsorption in the renal proximal tubules.

The possible effects of SGLT2 inhibitors on various liver complications and the potential underlying mechanisms have been discussed.

If SGLT2i is an important factor influencing type 2 diabetes mellitus, nephrological, and hepatological conditions, a summary Table or Figure may facilitate a clearer understanding of the risk contributions of SGLT2i to these conditions, as well as the effect size of SGLT2 inhibitors.

Similar information regarding GLP-1ARs is also suggested to be provided in a summary Table or Figure.

3.

To ensure the link between hepatic disorders and CKD is not overlooked in the comprehensive management of patients with type 2 diabetes mellitus, a schematic presentation of the factors contributing to the impact of uncertainty on type 2 diabetes, nephrological, and hepatological conditions is suggested.

Author Response

We thank the reviewer for all his suggestions. We appreciate the insightful comments made by the reviewer. In response, we made our best efforts to improve the manuscript. Please see the attached file "Reviewer 2" for our replies. All corrections in the text are written in red and highlighted in yellow.

Reviewer 3 Report

Comments and Suggestions for Authors

In the manuscript submitted for review, the authors have reviewed the literature on the characterization of risk factors and prevention of type 2 diabetes, focusing on nephrological and hepatological aspects. The topic addressed by the authors is important and timely. The manuscript is interestingly written and presented in a well-organized manner. The literature cited is mostly relevant publications. The figures are appropriate. The conclusions are consistent with the evidence and arguments presented. To attract the attention of the potential reader, I suggest some improvements:

1. Please develop a table summarizing the most important recommendations for clinicians.

2. Please develop a table with information on the progress of knowledge on new antidiabetic molecules and currently ongoing clinical trials in this area of expertise.

3. It is recommended to provide information on the software used to prepare the schemes.

4. ‘References’ should be prepared by the instructions for authors.

Author Response

We thank the reviewer for all his suggestions. We appreciate the insightful comments made by the reviewer. In response, we made our best efforts to improve the manuscript. Please see the attached file "Reviewer 3" for our replies.  All corrections in the text are written in red and highlighted in yellow.

Round 2

Reviewer 2 Report

Comments and Suggestions for Authors

All the concerns have been answered.

Reviewer 3 Report

Comments and Suggestions for Authors

The authors have revised their manuscript and responded to all suggestions. I recommend the article for further proceedings.